# The Human Gastric Juice: A Promising Source for Gastric Cancer Biomarkers

**DOI:** 10.3390/ijms24119131

**Published:** 2023-05-23

**Authors:** Nayra Felípez, Sheyla Montori, Naroa Mendizuri, Joan Llach, Pedro G. Delgado, Leticia Moreira, Enrique Santamaría, Joaquín Fernández-Irigoyen, Eduardo Albéniz

**Affiliations:** 1Gastrointestinal Endoscopy Research Unit, Navarrabiomed, Hospital Universitario de Navarra (HUN), Navarra Institute for Health Research (IdiSNA), Universidad Pública de Navarra (UPNA), 31008 Pamplona, Spain; nfelipev@navarra.es (N.F.); smontorp@navarra.es (S.M.); 2Clinical Neuroproteomics Unit, Proteomics Platform, Navarrabiomed, Hospitalario Universitario de Navarra (HUN), Navarra Institute for Health Research (IdiSNA), Universidad Pública de Navarra (UPNA), 31008 Pamplona, Spain; snmendizuri@gmail.com (N.M.); enrique.santamaria.martinez@navarra.es (E.S.); joaquin.fernandez.irigoyen@navarra.es (J.F.-I.); 3Department of Gastroenterology, Centro de Investigación Biomédica en Red en Enfermedades Hepáticas y Digestivas (CIBEREHD), IDIBAPS (Institut d’Investigacions Biomèdiques August Pi i Sunyer), 08036 Barcelona, Spain; jllachr@clinic.cat (J.L.); lmoreira@clinic.cat (L.M.); 4Facultat de Medicina i Ciències de la Salut, Universitat de Barcelona, 08036 Barcelona, Spain; 5Gastroenterology Department, Hospital de Mérida, 06800 Mérida, Spain; pgdg@gmail.com; 6Gastroenterology Department, Hospital Universitario de Navarra (HUN), Navarrabiomed, Navarra Institute for Health Research (IdiSNA), Universidad Pública de Navarra (UPNA), 31008 Pamplona, Spain

**Keywords:** gastric cancer, gastric juice, biomarker, early diagnosis, proteomics

## Abstract

Gastric cancer (GC) is a major public health problem worldwide, with high mortality rates due to late diagnosis and limited treatment options. Biomarker research is essential to improve the early detection of GC. Technological advances and research methodologies have improved diagnostic tools, identifying several potential biomarkers for GC, including microRNA, DNA methylation markers, and protein-based biomarkers. Although most studies have focused on identifying biomarkers in biofluids, the low specificity of these markers has limited their use in clinical practice. This is because many cancers share similar alterations and biomarkers, so obtaining them from the site of disease origin could yield more specific results. As a result, recent research efforts have shifted towards exploring gastric juice (GJ) as an alternative source for biomarker identification. Since GJ is a waste product during a gastroscopic examination, it could provide a “liquid biopsy” enriched with disease-specific biomarkers generated directly at the damaged site. Furthermore, as it contains secretions from the stomach lining, it could reflect changes associated with the developmental stage of GC. This narrative review describes some potential biomarkers for gastric cancer screening identified in gastric juice.

## 1. Introduction

Gastric cancer (GC) incidence is declining in many countries, but it remains one of the most common cancers globally. With 1 million new cases and 768,793 deaths in 2020, GC ranks fifth among the most common malignancies and fourth as a cause of cancer death (behind lung, colorectal, and liver cancer). Its incidence is higher in males than in females (7.1% vs. 4.0%), as well as its mortality (9.1% vs. 6.0%) [1]. The incidence of GC varies significantly around the world. The highest incidence is found in East Asia (particularly in Japan and Mongolia) and Eastern Europe as opposed to North America and North Europe [1].

GC is not a homogenous disease as it can be divided into two topographical subsets. These two types differ depending on their location. The Non-Cardia GCs (NCGCs) are those occurring in the distal stomach, and the Cardia GCs (CGCs) are those occurring where the stomach adjoins the esophageal-gastric junction [2]. The pathogenesis of both NCGC and CGC is multifactorial and involves several genetic, epigenetic, and environmental factors. *Helicobacter pylori* (*H. pylori*) is the leading risk factor for the development of NCGC. It has been linked to 90% of NCGC cases worldwide, while CGC has been connected mainly to obesity and gastroesophageal reflux [3,4]. NCGC and CGC share additional risk factors such as dietary habits (consumption of large amounts of salt-preserved foods, processed meats, and a low intake of fruits and vegetables), alcohol intake, smoking, or pathogenic infections with *H. pylori* or *Epstein Barr virus* (EBV) [5]. Although most GCs are sporadic, it has been estimated that around 10% of GC cases are inherited, and 1–3% of those affected by the disease have genetic mutations [6]. At least three significant conditions are associated with an increased risk of GC: hereditary diffuse gastric cancer (HDGC), gastric adenocarcinoma and proximal polyposis of the stomach (GAPPS), and familial intestinal gastric cancer (FIGC) [7].

In 2017–2021, the overall age-standardized 5-year relative survival rate for GC was 42.9% [8]. Despite declining incidence in many countries, GC remains a major public health concern. Avoiding risk factors and facilitating early detection through screening are key strategies for reducing the burden of this disease.

Currently, two main approaches to prevent GC are considered standard. As part of primary prevention, lifestyle and dietary changes are recommended, including avoiding smoking, restricting alcohol consumption, eating a balanced diet rich in fruits, vegetables, and fiber, reducing processed meat and salt-preserved foods, exercising regularly to maintain a healthy weight, and reducing the prevalence of *H. pylori* infections. In addition to screening for GC, upper gastrointestinal endoscopy is considered the gold standard for diagnosing or treating the disease using minimally invasive methods such as endoscopic mucosal resections and endoscopic submucosal dissections [9]. At present, gastric juice (GJ) is typically discarded during upper gastrointestinal endoscopy. However, utilizing GJ as a biomarker reservoir could enhance endoscopic screening accuracy and contribute to monitoring and surveillance of potential GC patients. By collecting and analyzing GJ, healthcare professionals may be able to detect missed lesions or abnormalities that may have been overlooked during endoscopic examination. This additional information could lead to earlier detection and intervention [10].

Moreover, monitoring changes in the composition or biomarkers present in GJ over time could provide valuable insights into disease progression, response to treatment, and overall patient management. This monitoring may enable more personalized and targeted approaches to patient care [10]. Overall, exploring this area could have significant implications for improving the accuracy and effectiveness of endoscopic screening, potentially improving patient outcomes.

This article addresses the challenges associated with late diagnosis and limited treatment options of GC by exploring the potential of GJ as a rich source of disease-specific biomarkers. It aims to highlight the significance of biomarker research in improving the early detection of GC and to discuss the advancements in technology and research methodologies for identifying potential proteomic biomarkers. By focusing on GJ as a “liquid biopsy” obtained during gastroscopic examinations, the article aims to underscore its potential for providing biomarkers directly from the site of GC origin and reflecting changes associated with the developmental stage of the disease. Furthermore, the article provides a narrative review of the identified biomarkers in GJ, their diagnostic value, and potential clinical applications, aiming to guide future research efforts and contribute to improved GC screening and patient outcomes.

## 2. Molecular Classification of Gastric Cancer

Traditionally, GC has been classified based on its histology. The most popular system is Lauren’s classification, which divides gastric adenocarcinoma into two main categories: diffuse and intestinal [11]. Diffuse GC (DGC) features a widespread infiltration of the stomach wall by cancerous cells with minimal or no glandular formation. At the same time, intestinal-type tumors (IGC) are characterized by glandular structures lined with malignant cells [12]. On the other hand, the World Health Organization (WHO) categorizes GC into four subtypes: tubular, mucinous, papillary, and poorly cohesive [11].

Recent advantages of next-generation sequencing (NGS) have substantially expanded the molecular understanding of tumor heterogeneity. The Cancer Genome Atlas (TCGA) established a novel classification system for gastric cancer based on molecular features, which could help identify altered pathways. The TCGA system divides gastric cancer into four subtypes: EBV, Microsatellite Instability (MSI), Genomically Stable (GS), and Chromosomal Instability (CIN) [13]. Each subtype is associated with distinct molecular and clinical features, providing insight into the underlying biology of GC [13] (Figure 1a).

In 2015, the Asian Cancer Research Group (ACRG) developed a new categorization based on gene expression data from 300 primary gastric tumors and identified four distinct subtypes: microsatellite stability/epithelial-mesenchymal transition (MSS/EMT), microsatellite-unstable tumors (MSI), microsatellite stable TP53-active (MSS/TP53+) and microsatellite stable TP53-inactive (MSS/TP53-) [14]. The ACRG categorization system included more detailed patient outcome data than TCGA, making it difficult to determine whether these subtypes have prognostic significance [15] (Figure 1b).

In summary, the TCGA and ACRG reports offer invaluable knowledge about the molecular subtypes of GC. The TCGA report classified four distinct molecular subtypes of GC and highlighted the importance of a molecular approach to diagnose and treat the disease. Additionally, the ACRG report provided further details on patient outcomes and recurrence patterns, which could be advantageous for future studies. Together, these two reports provide an expansive overview of GC at a molecular level that can be utilized to direct clinical decisions and guide future research endeavors.

## 3. Gastric Fluid Physiopathology: Risk Factors of GC

The digestive system is essential for both the intake of food and the detoxification of bodily wastes. The human gut contains microorganisms responsible for several biological processes but can also cause disease. The increased pH levels of the gastric juice and, therefore, the stomach environment makes it possible for harmful pathogenic bacteria to colonize the gut. This new environment allows bacteria to release nitrites and nitroso-compounds, ultimately causing an atrophy-metaplasia-dysplasia-carcinoma progression in the cells [16].

### 3.1. Helicobacter Pylori

Helicobacter Pylori is the most frequent bacteria to colonize the stomach. It has been classified as a class-I carcinogen. *H. pylori* participates in the multistage carcinogenic process of GC, known as the Correa Cascade, in which the gastric mucosa develops into gastric adenocarcinoma [17]. This bacterium is most prevalent in developing countries, found in 85–95% of the population [18]. On the other hand, due to better sanitation practices [19], the incidence of *H. pylori* infection is lower in developed countries, such as those with high-income or upper-middle-income countries, where it is only found in 30–50% of the population [18]. It is estimated that only a small fraction, less than 5% of those infected, will develop cancer, which may be due to differences in bacterial genetics, age of infection, high historical rate of transmission, and environmental cofactors [1,18,19].

Over the years, mixed results have suggested that *H. pylori* infection may be associated with CGC, but the evidence is inconclusive [20]. This is because studies conducted in Western countries have shown no correlation or even a negative correlation between CGC and infection of this bacteria. However, evidence exists of an increased risk of CGC among those infected in high-risk areas [21]. For example, a systematic review and meta-analysis [22] of 30 studies conducted in both high-risk and low/middle-risk areas found no correlation between *H. pylori* infection and CGC. However, when the data were divided into groups based on GC incidence, a positive association was observed in high-risk populations with a relative risk (RR) of 1.98 (95% CI 1.38–2.83). Additionally, a recent prospective case-cohort study [20] in the Chinese population revealed that this bacterium is indeed a risk factor for CGC with a seroprevalence of 92.2% (89.7–94.7) and adjusted hazard ratios (HRs) of 3.06 (1.54–6.10).

*H. pylori* infection can lead to chronic inflammation and the formation of atrophic gastritis [23]. The proteins CagA and VacA are two essential virulence factors produced by certain strains of *H. pylori* that can increase the risk of developing GC [15]. Both of these proteins contribute to the development of GC through their effects on epithelial cells in the stomach, leading to changes in cell signaling pathways which can ultimately lead to cancerous transformation [24]. They can also induce cytokine production, such as interleukin-1β, leading to inflammation and promoting carcinogenesis through sustained ROS production [25].

### 3.2. Epstein-Barr Virus

Nearly 10% of GC cases are associated with EBV, the leading cause of acute infectious mononucleosis [26]. This pathogen was the first to be identified as carcinogenic in humans, and it is one of the primary risk factors for Hodgkin’s lymphoma or nasopharyngeal carcinoma [27]. EBV has a direct transforming effect on host cells by expressing its regulatory genes. This allows EBV to replicate within the infected cells, causing excessive growth and potentially leading to malignant transformation [28]. DNA methylation and CpG island hypermethylation are also characteristics of EBV. These alterations can lead to silencing various genes, particularly tumor suppressor genes. The effects of this process play an essential role in cancer development by influencing expression levels of important oncogenes while restraining tumor suppressor pathways [29].

## 4. The Regulation of the Production of Gastric Juice

The release of HCl is triggered by the arrival of food in the stomach. The increased production of GJ helps to break down food and absorb nutrients. Endocrine, paracrine, and neural pathways regulate the secretion of gastric acid. Stomach acid production is primarily activated by histamine, gastrin, and acetylcholine (Ach), whereas somatostatin directly inhibits this process [30] (Figure 2). The role of stimulant hormones relies on binding to receptors to the parietal cells, which triggers a cascade of events, such as stimulating the proton pumps that actively transport H+ ions, leading to increased HCl secretion [31].

On the other hand, inhibitory hormones such as somatostatin, released in response to food intake, act on parietal cells to decrease acid production. Other hormones, such as cholecystokinin (CCK) or secretin, regulate acid production [32] (Figure 2). These hormones stimulate the release of bicarbonate ions from the pancreas into the duodenum, which neutralize the acid and protect the mucosal lining from damage [33].

## 5. Gastric Juice Collection and Biomarkers in Gastric Cancer

Currently, GC biomarkers used in daily routine are not specific or sensitive enough, and most are obtained invasively. The appearance of new non-invasive biomarkers capable of diagnosing early-stage GC is a promising strategy [34]. GJ has become increasingly important and noteworthy in recent years due to its potential for yielding GC-associated biomarkers. Since it is generated in the same place as the disease originates, it stands to reason that this body fluid may contain more substances from tumor cells than other fluids.

GJ is a digestive fluid produced in the stomach. It is the first barrier between the digestive tract and pathogenic substances. It contains HCl, lipase, and pepsin [35]. The acidic pH of gastric juice is due to HCl secreted by the parietal cells, the corrosive effect of which is countered by glycoprotein mucins produced by the mucous cells [36]. A gastric aspirator suction device can collect GJ samples during oesophageal-gastro-duodenoscopy (OGD). This allows suctioning of gastric juice while avoiding contact with other organs or tissues in the digestive tract. Subsequently, GJ can be processed or stored at a temperature of −80 °C for future testing.

An expanded range of sample collection sources is necessary to enhance the screening of GC further and discover new markers since conventional serum biomarkers such as carcinoembryonic antigen (CEA) have yet to achieve high sensitivity and specificity [37]. The main downside of GJ is that it must be acquired through an endoscopic procedure, which is harder to obtain than other bodily fluids. The studies thus far seem promising when using samples like blood, but since many cancers share similar alterations and biomarkers, obtaining them from the site where the disease originates tends to yield more specific results (Table 1).

### 5.1. Non-Coding RNAs

Non-coding RNAs (ncRNAs) are broadly distributed throughout living organisms and cannot produce proteins. These can be broken down into three distinct sizes: small ncRNAs, including siRNAs, miRNAs, and piRNAs; mid-sized ncRNA; and long ncRNAs (lncRNA) [38].

#### 5.1.1. microRNAs

MicroRNAs (miRNAs) are small non-coding RNAs that regulate gene expression, and since they are frequently observed in other types of cancers, GJ miRNAs might offer more specific insights [37]. Several studies have reported that miRNAs could be promising biomarkers for the early detection and diagnosis of GC [39,40,41]. Zhang et al. conducted an experiment where they collected GJ from 141 patients, 42 of whom had GC (7 in the early stage and 35 advanced cases) [39]. The miR-421 levels were assessed in the specimens, and it was determined that those with GC had significantly lower miR-421 levels than those with benign diseases (*p* < 0.001). Furthermore, this research suggests a remarkable improvement in early GC detection when combining GJ miR-421 and juice CEA instead of just serum CEA alone (*p* = 0.029) [39]. Subsequent research focused on other miRNAs such as miR-129-1-3p/miR-129-2-3p [40], miR-106a, and miR-21 [41]. They all showed significantly lower expression levels in GC patients than those with benign gastric diseases. Combining these biomarkers is more effective for detecting the early stages of GC [40,41].

Similarly, in another cohort, Shao et al. collected GJ from 204 subjects, of which 62 had GC [42]. They evaluated the levels of miR-133a, and as mentioned above, it was downregulated. They also measured the correlation of expression of miR-133a in GC tissues and GJ. Pearson’s correlation test demonstrated a strong positive correlation (0.972, *p* < 0.0001) between the expression of miR-133a in GC tissues and GJ, suggesting that it could be used as a potential biomarker for diagnosing GC [42].

Thus, these results suggest the potential application of microRNAs as novel diagnostic markers for GC screening tests. Further research is needed to validate their utility as reliable biomarkers before they can be used clinically. Still, these molecules promise to improve our ability to detect GC earlier and more accurately than ever before.

#### 5.1.2. piRNAs

piRNAs are small ncRNAs that usually bind to a family of proteins called PIWI. This complex is essential in germline development and gametogenesis [43]. Little is known about this type of ncRNA; therefore, few studies have worked with piRNAs as possible biomarkers in cancer. In 2020, Zhou et al. conducted a study in China, collecting samples of 132 patients (66 with GC and 66 healthy) [44]. They aimed to determine piRNA-1245 as a possible candidate for a GC biomarker. The levels of piRNA-1245 were higher in GC patients (*p* < 0.0001) than in healthy ones. Additionally, their survival analysis revealed that individuals with high expression of piR-1245 had poorer overall survival (OS) and progression-free survival (PFS) rates than those with low expression of piR-1245 (*p* = 0.0152 and *p* = 0.013, respectively) [44].

#### 5.1.3. lncRNAs

lncRNAs have been the subject of extensive research. Although some studies have yielded promising results, either the sample sizes needed to be more significant for more definitive outcomes or the sensitivity rates were missing or less than 50%. Pang et al. conducted an experiment where they collected samples from 71 patients with GC, but only 17 samples of GJ were measured [45]. Linc00152 was upregulated in patients with GC (*p* < 0.002) [45]. Other lncRNAs were studied in the last years, such as AA174084 [46], RMRP [47], ABH11-AS1 [48], LINC00982 [49], H19 [50], and UCA1 [51]. They all share that they were significantly higher expressed in patients with GC, the favorable rates between early and advanced GC were >50%, and the results of combining lncRNAs and classic serum biomarkers were improved [46,47,48,49,50,51]. Also, it is worth mentioning that AA174084, RMRP, and LINC00982 levels were decreased in GC tissues; however, they were higher in GJ in patients with GC, which could be explained by their role during gastric carcinogenesis [46,47,49].

### 5.2. DNA

Hypermethylation of CpG islands and the consequent tumor-associated and tumor suppressor genes silencing is the key epigenetic feature of GC, making it possible to identify potential biomarkers [52]. Yamamoto et al. conducted a study with 20 GJ samples from patients with GC [53]. They found that hypermethylation of the BARHL2 gene was related to the early stage of GC in patients who had not yet undergone endoscopic resection. After the procedure, methylation levels dropped considerably. These results imply that BARHL2 methylation could be used to detect residual cancer post-endoscopic resection and potentially forecast relapses. Additionally, they did not find any significant relationship between BARHL2 methylation and atrophy or *H. pylori* infection [53]. Nonetheless, more research must explore the clinical significance and mechanism of BARHL2 hypermethylation concerning GC.

### 5.3. Proteins

In GC research, little is known about the relationship between protein expression levels and GJ. Most studies focused on serum levels, resulting in unspecific biomarkers. However, a proteomic analysis discovered significant differences in protein levels between healthy participants and patients with symptomatic gastric diseases. A MALDI-TOF mass spectrometer was used in studies where α1-antitrypsin or its precursor was one of the most significant novel biomarkers. Hsu et al. [54] conducted a study with 31 GC patients. Protein concentrations were higher in the GC group than in the healthy group. The study also demonstrated differences in protein components, with a frequency of 93% of the specific α1-Antitrypsin precursor band vs. 6% in healthy donors [54]. A previous study by Lee et al. [55] also revealed an association between α1-Antitrypsin and GC.

This review repeatedly emphasizes that GJ is a valuable source of biomarkers in GC research. Early detection techniques with a low level of invasiveness are the main goal of GC researchers. Over the years, proteomic techniques have advanced, and besides mass spectrometry, other novel techniques have become established. Therefore, as one of the most promising fields in the search for new biomarkers, we have detailed in more depth some of the most relevant studies in proteomics and some of the most innovative techniques.

**Table 1 ijms-24-09131-t001:** Gastric juice biomarkers for early detection and prognosis of gastric cancer.

Ref. No.	Type	Biomarkers	GC/nonGC Patients	Expression Level *	AUC (95% CI)	Sensitivity (%)	Specificity (%)
[39]	miRNA	miR-421	42/99	Downregulated (*p* < 0.001)	0.77 (0.68–0.85)	71.40	71.70
[40]	miRNA	miR-129-1-3p and miR-129-2-3p	42/99	Downregulated (*p* < 0.001)	0.66 (0.56–0.76)	68.70	71.90
[41]	miRNA	miR-106a	42/99	Downregulated (*p* < 0.001)	0.87 (0.80–0.95)	73.80	89.30
[41,42]	miRNA	miR-21	42/99	Downregulated (*p* < 0.001)	0.97 (0.94–1)	85.70	97.80
[42]	miRNA	miR-133a	62/142	Downregulated (*p* < 0.001)	0.91 (0.86–0.96)	85.90	84.80
[44]	piRNA	piR-1245	66/66	Upregulated (*p* < 0.0001)	0.89 (0.83–0.94)	90.90	74.20
[45]	lncRNA	Linc00152	17/16	Upregulated (*p* = 0.002)	-	-	-
[46]	lncRNA	AA174084	39/92	Upregulated (*p* < 0.01)	0.85 (0.78–0.92)	46.00	93.00
[47]	lncRNA	RMRP	39/92	Upregulated (*p* < 0.01)	0.70 (0.59–0.81)	56.40	75.40
[48]	lncRNA	ABHD11-AS1	39/92	Upregulated (*p* < 0.033)	0.65 (0.54–0.77)	41.00	93.40
[49]	lncRNA	LINC00982	27/27	Upregulated (*p* < 0.026)	-	-	-
[50]	lncRNA	H19	33/23	Upregulated (*p* < 0.034)	-	-	-
[51]	lncRNA	UCA1	26/23	Upregulated (*p* < 0.016)	-	-	-
[53]	DNA	BARHL2	20/10	-	0.92	90.00	100.00
[54]	Protein	α1-Antitrypsin	31/120	-	-	-	-
[55]			30/5	-	-	-	-

GC: gastric cancer; AUC: area under the curve; Ref.: reference; -: no data available. * compared with adjacent non-GC patients.

## 6. Deployment of Proteomics in Gastric Juice

In recent years, proteomics has emerged as a powerful toolbox to characterize the molecular mechanisms behind GC progression and identify potential biomarkers and therapeutic targets [56,57,58]. More recently, multi-proteomic analyses have been used to dissect GC heterogeneity, characterizing three subtypes of DGC and IGC, and advancing a more comprehensive understanding of precision oncology [59]. Considering the GJ as a waste product during a gastroscopic examination, GJ could provide a “molecular biopsy” enriched in disease-specific protein biomarkers [60] generated directly at the damaged site. However, an essential preanalytical variable is the GJ pH level [61] due to its impact on protease activity in the stomach. Depending on the pH level of the GJ, stomachal proteins may be subjected to acid hydrolysis, alkaline denaturation, or peptide digestion, dramatically impacting the dynamics of the GJ proteome.

Moreover, it is important to note that wide variations in GJ pH levels exist across the population [62,63], changing with age [64,65], infections [66], and oncologic processes [63], being a relevant parameter to avoid confounding results and erratic outcomes in biomarker discovery or monitoring studies. The number of mass spectrometry-based proteomic studies used to generate GJ protein profiles is generally low (Table 2). Most are unbiased, directly measuring protein identity and abundance by measuring peptides between healthy conditions and GC to improve the biomarker discovery pipelines [54,55,67,68,69]. Large cohorts can be monitored at a depth of hundreds of GJ proteins using isobaric labeling (TMT/iTRAQ) or data-independent acquisition approaches (DIA-SWATH). Common limitations are due to the large dynamic range of protein concentrations in GJ and the accumulative problem of missing values when processing large patient cohorts [70]. Complementary to mass spectrometry, several protein measurement technologies are potentially available to perform clinical proteomic pipelines. These innovative approaches are mainly based on multiplexed antibodies (proximity extension assay (PEA) technology from Olink^®^, Watertown, MA, USA) and multiplexed nucleic acid aptamers (SomaScan^®^aptamer-based technology from SomaLogic, Inc.; Boulder, CO, USA). PEA applies a sandwich antibody-based system where the capture and detection of antibodies are conjugated to a complementary oligonucleotide probe pair, the levels of which are measured by quantitative PCR or next-generation sequencing, providing relative protein abundance levels [71]. To our knowledge, Olink^®^ technology has not been used yet in the context of GJ. However, different reports point out the utility of using serum as a biological sample in predicting responses to preoperative chemotherapy for GC [72] and characterizing novel immunological GC biomarkers [73,74,75,76].

On the other hand, SomaScan^®^ is based on modified DNA aptamers with slow off-rate binding kinetics to monitor relative protein levels in a multiplex format [77]. This mechanism can generate multiple aptamers for each targeted protein and be incorporated in the microarray-based readout of relative fluorescence intensity. This technology is not widely used at the level of GJ. However, SomaScan^®^ has been successfully applied in multiple gastric contexts [78,79,80,81,82]. It has been pointed out that both proteomics approaches theoretically suffer less from dynamic range and missing value problems compared to mass-spectrometry. However, protein modifications and off-target binding effects may impact their specificity and accuracy, and specific reagents should be optimized for each target protein [83].

Interestingly, both approaches have been successfully applied in parallel in different biofluids [84,85,86,87]. However, to our knowledge, no studies have compared these affinity-based methodologies using GJ. Considering that Olink^®^ and SomaScan^®^ technologies can detect up to 3000 and 7000 proteins, respectively, applying both approaches in a minimal quantity of GJ will push the development of novel biomarker discovery pipelines necessary to implement the personalized management of GC.

**Table 2 ijms-24-09131-t002:** MS-proteomic studies.

Ref. No.	Study Subjects	Analytical Platform	Findings
[54]	healthy subjects (n = 120)	2D-electrophoresis MALDI-TOF MS LC-MS/MS	Alpha-1-antitrypsin precursor as a novel gastric juice biomarker for gastric cancer and ulcer.
	gastric ulcer (n = 39)		
	duodenal ulcer (n = 38)		
	gastric cancer (n = 31)		
[55]	99 individuals	2D-electrophoresis MALDI-TOF MS LC-MS/MS	Strong association of a high level of alpha-1-antitrypsin in gastric juice with gastric cancer
	gastric cancer (n = 30)		
	gastrititis (n = 56)		
	Others (n = 13)		
[61]	benign gastric conditions (n = 170)	2D-electrophoresis Multidimensional LC-MS/MS	Distinct protein profiles for acidic and neutral samples, highlighting pH effects on protein composition
[67]	gastric cancer (n = 19)	SELDI-TOF MS	60 proteomic features were up-regulated, and 46 were down-regulated in gastric cancer samples.
	benign gastritis (n = 36)		
[68]	normal (n = 106)	MALDI TOF-TOF ESI-MS/MS	Five peptides for gastric cancer diagnosis with high sensitivity and specificity (derived from pepsinogen, leucine zipper protein, albumin and a-1-antitrypsin fragment)
	duodenal ulcer (n = 38)		
	gastric ulcer (n = 38)		
	gastric cancer (n = 34)		
[88]	chronic gastritis (n = 6)	MALDI TOF-TOF	First report of the proteome of human gastric juice with gastritis background (327 proteins)
[89]	gastric cancer (n = 70)	iTRAQ and SWATH LC-MS/MS	A biomarker panel scoring matrix for early GC detection (diagnostic sensitivity of 95.7%)
	benign gastritits (n = 17)		

## 7. Conclusions and Future Perspectives

GC is a significant public health concern worldwide, with high mortality rates due to late diagnosis and limited treatment options. Despite ongoing efforts to find biomarkers for early diagnosis, much remains to be learned about how they can be used in clinical practice. Nonetheless, technological advancements and research methodologies instill optimism for new diagnostic tools that could enhance patient outcomes. Recent advances in biomarker research have identified several potential biomarkers for GC, including microRNAs, DNA methylation markers, and protein-based biomarkers. The search for biomarkers has been a challenge. Although most studies have focused on identifying biomarkers in biofluids like blood, the low specificity of these markers has hindered their use in clinical practice. As a result, recent research efforts have shifted towards exploring GJ as an alternative source for biomarker identification. GJ is a potential source for biomarker research due to its proximity to the tumor site. It contains secretions from the stomach lining that may reflect changes associated with GC development. Recently, proteomics has emerged as a powerful toolbox to characterize the molecular mechanisms behind GC progression and identify potential biomarkers and therapeutic targets [56,57,58]. However, the mass spectrometry-based proteomic studies used to generate GJ protein profiles are generally low [54,55,61,67,68,88,89]. Latest-generation protein measurement technologies, such as Olink^®^ and SomaScan^®^, are potentially available to perform clinical proteomic pipelines. Considering these technologies can detect up to 7000 proteins, their application in minimal GJ will allow clinicians to improve diagnostic accuracy and develop more effective treatments.

In the end, identifying new biomarkers is crucial for the early detection and diagnosis of the disease, which can significantly improve the patient’s chances of survival and quality of life. It is necessary to develop reliable biomarkers that can accurately detect GC and its premalignant lesions at earlier stages to improve patient outcomes. Biomarker-based screening and monitoring of high-risk populations could also reduce the burden of the disease by identifying individuals who may benefit from early intervention. Additionally, biomarker research may help identify new therapeutic targets and guide the development of personalized treatments for GC patients. Therefore, continued research in this area is essential to combat this disease.

## Figures and Tables

**Figure 1 ijms-24-09131-f001:**
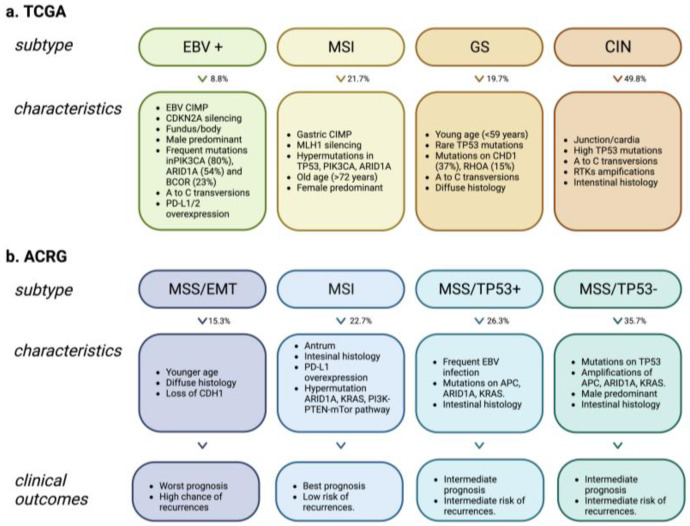
Molecular classification of gastric cancer. Characteristics associated with each subtype according to TCGA and ACRG studies. (**a**). The Cancer Genome Atlas (TCGA) stratified gastric cancer into four subgroups. EBV+: Epstein Barr virus-positive; MSI: Microsatellite instability; GS: Genomically stable; CIN: Chromosomal instability. (**b**). The Asian Cancer Research Group (ACRG) classified gastric cancer into four subtypes. MSS/EMT: Microsatellite stability/epithelial-mesenchymal transition; MSI: Microsatellite-unstable tumors; MSS/TP53+: Microsatellite stable TP53-active; MSS/TP53-: Microsatellite stable TP53-inactive. Created with BioRender.com.

**Figure 2 ijms-24-09131-f002:**
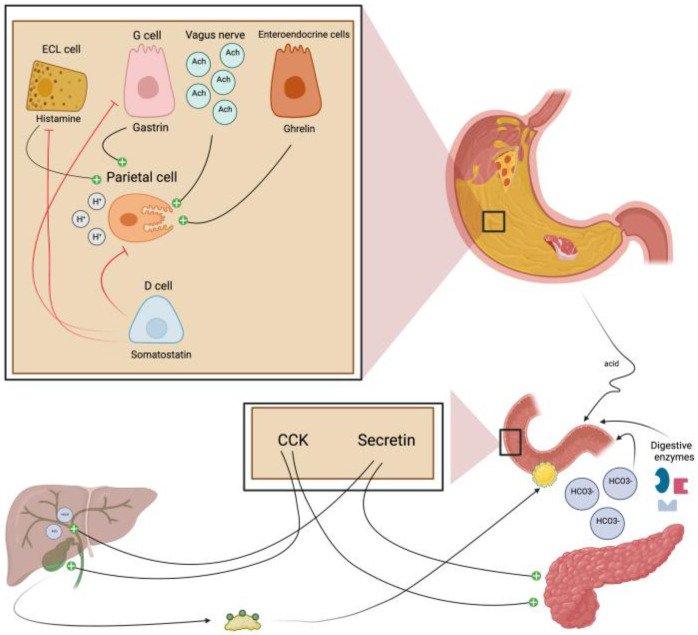
Main regulators of gastric juice production. GJ secretion is tightly regulated by a series of mechanisms that involve both neural and hormonal pathways. Histamine (released by ECL cells), gastrin (released by G-cells), and acetylcholine (released from postganglionic neurons of the vagus nerve) increase the production of gastric acid. GJ secretion can also be inhibited through several different mechanisms. Enterogastrone hormones such as secretin and CCK are released from enteroendocrine cells throughout the small intestine after food ingestion, inhibiting gastric emptying and reducing GJ production. These hormones can also stimulate the release of bile and pancreatic juices, allowing not only better digestion of fatty acids in the small intestine but also countering the acidic effect of GJ that could reach the duodenum. Finally, somatostatin produced by D-cells decreases gastric secretion rates due to its effects on inhibiting the further release of other stimulatory hormones, such as histamine. ECL cell: enterochromaffin-like cell; G cell: gastrin cell; Ach: acetylcholine; D cell: delta cell, CCK: cholecystokinin; HCO3- bicarbonate. Created with BioRender.com.

## Data Availability

No new data were created or analyzed in this study. Data sharing is not applicable to this article.

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
