# Peer review of "The Human Gastric Juice: A Promising Source for Gastric Cancer Biomarkers"

_ijms, 2023, doi:10.3390/ijms24119131_

Round 1

Reviewer 1 Report

The authors Nayra Felípez and their colleagues made an interesting article on human gastric juice: a promising source for gastric cancer biomarkers.

The significance of identifying biomarkers in gastric cancer was amply demonstrated in the article. Overall, the manuscript is clear and well-written. This paper is critically essential and timely for the subject area.

Minor remarks:

1. The manuscript contains a few typos; please fix them.

2. Create a visual representation of all the biomarkers that have been connected to gastric cancer to this point. I recommend the manuscript for acceptance with minor corrections.

I recommend the manuscript for acceptance with minor corrections.

Reviewer 2 Report

In this manuscript the authors overviewed and systematized existing to date potential GC biomarkers identified in gastric juice. The subject of the reviewed manuscript seems to be relevant and topical. However, important concerns shown below should be addressed to the authors before publication.

Lines 74-75: 'role of gastric juice (GJ) as a new potential biomarker'

Not gastric juice, but its components. GJ couldn't be considered as biomarker. Or 'a new potential biomarker SOURCE for gastric cancer screening'

Table 1:

lnRNA instead of lncRNA

piRNA - not found in the abbreviations list (as some other abbreviations not mentioned here).

The authors undeservedly ignored the miRNA families that regulate the activity of EMT driving transcription factors (Zeb1/2, Snail/Slug, Twist etc.) playing crucial role in tumor metastasis.

The miR-21 microRNA shown to be downregulated in GC (Table 1) is known as oncogenic one and up-regulated by Zeb1.

Prior to applying PEA & SomaScan, the set of GC protein oncomarkers should be defined, and the MS-based proteomic approaches seem to be most convenient here.

Abstract exceeds the allowed 200-word limit.

References: do not meet the journal's “Instructions for Authors”.

And finally, authors need to perform a thorough spell check of the manuscript as there are numerous mistakes (often curious ones - f.ex., in Line 285: MALDI-TOP) throughout the text and negligence in manuscript preparation, so the manuscript must be improved to meet the criteria established in the journal's “Instructions for Authors”.

Authors need to perform a thorough spell check of the manuscript as there are numerous mistakes (often curious ones - f.ex., in Line 285: MALDI-TOP) throughout the text and negligence in manuscript preparation, so the manuscript must be improved to meet the criteria established in the journal's “Instructions for Authors”.

Reviewer 3 Report

This manuscript has reviewed the published literature describing gastric cancer biomarker screening in gastric juice. The article is very well written; English language is excellent, with very few grammatical errors and generally good referencing. The length of the review is appropriate, and most relevant topics within this field have been covered. The authors could consider the following comments. Some minor grammatical errors are also listed as these may not he detected in final editorial review.

1. I feel that the overall topic of the review could be more clearly stated. ‘Biomarkers’ is a very broad term and in the context of a cancer, such as gastric cancer, it can apply to diagnosis, subtyping, prognostication, response to treatment, recurrence, discovery, amongst other subjects. The breadth of the review in this context is not clearly stated (what will and won’t be covered) and as such, the goals of the review are not well-defined. A suggestion would be to more decisively place the topic of the review in the context of the field as this would assist readers to better understand its purpose.

2. In line with the previous comment, the majority of the review appears to be focussed on gastric cancer diagnosis, however, the authors have not described current methods of gastric cancer diagnosis, where these may be lacking, and how biomarker development may ameliorate this. Because both gastric biopsy and sampling of gastric juice for biomarker analysis both involve endoscopy, and because tissue pathology (i.e. biopsy) is required for precise gastric cancer subtyping / clinical management planning, there needs to be a justification for the addition of further biomarker tests. What would the inclusion of these tests add in terms of diagnostic sensitivity, specificity or accuracy (for example)? These descriptions do not need to be comprehensive, but they will clarify the purpose of the review and its importance in the field.

3. As this is a review, the authors could re-check their list of references as several relevant references and prior reviews appear to be missing (e.g. DOI: 10.1002/ueg2.12328).

Minor errors (this list is not comprehensive)

1. Line 199-200: ‘the corrosive effects of which are’

2. Line 285: MALDI-TOF

3. Line 313: peptide digestion

4. Line 337: off-rate

5. Line 350: up to 3,000 and 7,000 proteins

6. Line 360: ‘remains to be learned’

7. Line 374: Latest generation

Round 2

Reviewer 2 Report

See the attachment. The reviewer's remarks made in blue.

The quality of English has improved if compared with previous version.
